# Moral Reasoning about Aggressive Behavior in Relation to Type of Aggression, Age and Gender in South Korean Pupils

**DOI:** 10.3390/ijerph18052288

**Published:** 2021-02-25

**Authors:** Seung-Ha Lee, Peter K. Smith, Claire P. Monks

**Affiliations:** 1Department of Early Childhood Education, Chung-Ang University, 84 Heukseok-ro, Dongjak-gu, Seoul 06974, Korea; 2Goldsmiths College, University of London, London SE14 6NW, UK; p.smith@gold.ac.uk; 3Institute for Lifecourse Development, School of Human Sciences, University of Greenwich, London SE10 9LS, UK; c.p.monks@greenwich.ac.uk

**Keywords:** moral reasoning, aggression, South Korea, children

## Abstract

Studies of moral reasoning in relation to aggressive behaviors have paid limited attention to different types of aggression, and have mainly been conducted in Western societies. We describe findings from a study of 157 children, aged 6 or 11 years, from two schools in South Korea. Using a cartoon scenario methodology, we assessed moral reasoning about eight types of aggression: verbal, physical individual, physical group, social exclusion, rumor spreading, breaking one’s belongings, sending a nasty text via mobile phone, and sending a nasty message/email via computer. Four aspects of moral reasoning were assessed: moral judgment, harmfulness, reason for judgment, and causal responsibility. Many significant differences by type of aggression were found, especially for social exclusion (seen as less wrong and harmful, and more the victim’s responsibility), physical group aggression (seen as more wrong or harmful, and a matter of fairness, especially in older children and boys), and cyber aggression (seen more as the aggressor’s responsibility). Older children gave more reasons based on welfare, and fewer “don’t know” responses for reasons and attributions. Gender differences were relatively few, but girls did make more use of welfare in the moral reasoning domain. Findings are discussed in relation to previous research and the cultural context in South Korea.

## 1. Introduction

Moral reasoning refers to the thinking processes employed when deciding whether a behavior is morally acceptable [1]. It covers various aspects, such as whether any harm is thought to be done by a behavior, whether the behavior is right or wrong (moral judgment), reasons given for the judgment, and causal attributions (why this behavior happened).

By definition, aggressive behavior has an intent to cause harm, and invites moral judgment about whether it is justified. Moral reasoning will be one part of the process by which we can understand why some individuals do or do not engage in such behavior [2]. Bullying behavior, also involving repetition and an imbalance of power [3], can be seen as an infringement of rights with an even stronger imperative for moral judgment considerations [4].

Aggression and bullying have been studied from the early years of schooling. However, young children have difficulty distinguishing between aggression and bullying, and indeed the concept of bullying has been questioned in preschool children, with a case being made that the term “unjustified aggression” is the best descriptor at this age [5]. Interviewing four to eight year olds in Australian early education settings, Ey, Walker, and Spears [6] found that children confused bullying with aggressive-only behavior. Nevertheless, these authors found that these children “drew on moral reasoning perspectives” in describing their understanding of scenarios presented to them.

Moral reasoning can involve an initial judgment about the harmfulness and wrongness of a behavior, but can also include reasons for why the behavior happened, and who was responsible. The kinds of reasons given were examined by Davidson, Turiel, and Black [7]; they examined moral reasoning in samples of 6, 8, and 10 year olds from California, USA. They presented a range of moral and conventional transgressions in short narrative scenarios, and asked children about reasons for these. They content analyzed responses into nine categories, organized into domains of “moral”, “prudential”, “social conventional”, and “personal”, based on the social domain approach [8]. Children can also have varying perceptions about causal attribution of responsibility: Smith et al. [9] examined such attributions among 13–16 year old pupils in UK schools, and classified them into victim-related reasons; relationship between the bully and victim; bully-related reasons and no reasons/don’t know.

There is some evidence that moral reasoning varies by the type of aggression being considered [10,11]. However, existing studies have used a very limited range of types of aggression, and most have only examined children in Western societies. In this paper, we examine aspects of moral reasoning about a range of aggressive behaviors in young children in South Korea. As a relatively strongly collectivist society, social exclusion as a type of aggression is very salient in South Korean schools [12,13], so moral reasoning about social exclusion is of particular interest. We next review relevant literature on age changes in moral reasoning, gender differences, variations by type of aggression, and cultural differences.

### 1.1. Age Changes

We know that aspects of moral reasoning vary by age; even in preschool, children aged 2.5 to 4 are increasingly able to distinguish moral from social conventional transgressions [14]. Davidson et al. [7] examined moral reasoning in samples of 6, 8, and 10 year olds from California, USA. They presented a range of moral and conventional transgressions in short narrative scenarios and asked children about their moral judgments and reasons or justifications for these. Reasons were content analyzed into nine categories, based on a social domain approach [8] with reasons for moral judgments organized into domains of “moral”, “prudential”, “social conventional”, and “personal”. Two of these showed significant age changes: younger children gave more reasons based on “appeal to authority”, whereas older children gave more reasons based on “personal choice”.

Scholes et al. [15] also used a social domain theory approach to examine moral reasoning around a scenario regarding rejection of an aggressive child, in a sample of 6–7 year olds in Australia. Over a one year period, rejection responses decreased, and moral justifications based around prosocial or empathic reasoning increased.

Helwig, Hildebrandt, and Turiel [16] examined 6 to 11 year old US children’s judgments about using verbal and physical aggression within a game context. They found that both younger and older children judged “calling peers stupid” or “pushing peers” as wrong even when they were rules in the game. However, this was more so among the older children than the younger children. Younger children (6–7 year olds) were more likely to ignore the perpetrator’s (aggressor’s) intention or the recipient’s (victim’s) perspective than older children (10–11 year olds).

### 1.2. Gender Differences

Gender differences in moral reasoning have been widely debated [17]. However, research has not found clear-cut gender differences in moral reasoning in children [18]. Neither Davidson et al. [7] or Scholes et al. [15] reported significant gender differences in their studies. In older samples, Park and Killen [19] found that girls more than boys thought that peer rejection was wrong, in samples of 10–13 year olds in the US and South Korea (irrespective of culture). Similarly, Thornberg and colleagues [20] reported that female students aged 10–13 years judged bullying as more wrong and explained their judgments using moral reasons more frequently than male students. Also in the US, Donenberg and Hoffman [21] reported that a tendency for moral reasoning around care-orientation in girls, but justice-orientation in boys, was relatively clear among 5^th^ and 6^th^ graders but became blurred among 10^th^ and 11^th^ graders.

However, gender differences may interact with the type of aggression being considered. In a study of US 4^th^ and 5^th^ graders, Murray-Close et al. [10] found that girls judged relational and physical aggression as more wrong and relational aggression as more harmful than boys; but girls more often used moral reasoning to judge relational compared to physical aggression. Boys were more likely than girls to identify aggression as a matter of social, conventional, and personal choice.

### 1.3. Moral Reasoning by Type of Aggression

Although they generally understand aggressive behavior is wrong and harmful [22], children may think about the level of wrongness or harmfulness differently, depending on the type of aggression. In US samples, Goldstein, Tisak, and Boxer [23] found that preschool pupils considered relationally aggressive responses as more acceptable than verbally or physically aggressive behavior. At 4^th^ and 5^th^ grade, Murray-Close et al. [10] reported that children generally viewed physical aggression as more wrong and more harmful than relational aggression.

Goldstein et al. [23] and Murray-Close et al. [10] examined three types of aggression: physical, verbal, and direct/relational (social exclusion). However, they did not examine other types of aggressive behavior such as instrumental (breaking one’s belongings), and indirect/relational (i.e., gossip, rumor spreading) [24]. In a slightly older sample of 12–13 year old children in the US, Goldstein and Tisak [25] included both social exclusion and gossiping (rumor spreading); gossip was seen as being as wrong as physical aggression, and more wrong than social exclusion; thus, although gossiping and excluding are both regarded as forms of relational aggression, the distinction between these two forms is important.

In addition, cyber aggression is also an important focus for research, even at younger ages [26]. Cyberbullying is different from traditional bullying in terms of the potential anonymity of cyberbullies, the rapid speed at which it can spread and thus the potential size of the audience it may reach, and that it may happen in secret [27]. These factors may mean that the bully or bystanders may be less likely to empathize with the victim, or perceive it as a moral issue. Wright et al. [11] assessed attributions for face-to-face and cyber victimization amongst adolescents (11–15 years) in six countries (China, Cyprus, Czech Republic, India, Japan, US). They distinguished self-blame, aggressor-blame, normative, joking, and conflict attributions. The first three attributions were more commonly used for public and face-to-face victimization than for private and cyber forms. Some gender and country differences were also reported, but did not show a very clear pattern.

### 1.4. Studies in South Korea

Bullying in South Korea is called wang-ta. In its etymology, the term literally indicates socially excluding or a socially excluded person. The person called wang-ta can be victimized by various types of aggressive behavior (physical, verbal, cyber), but predominantly in a social or relational way; nevertheless physical aggression can sometimes involve many children in a class attacking the victim [28]. Park and Killen [19] examined justifications for peer rejection in 10 and 13 year old children in South Korea, and the US. Cultural differences were small in this study. However, Lee, Smith, and Monks [13] found that in South Korea, it was more common to attribute responsibility of victimization to a socially excluded person than to a person victimized in other ways such as physical aggression. South Korea is a collectivist society (ranking 65^th^ highest out of 76 countries on collectivism, in Hofstede, Hofstede, and Minkov, [29]), and this may be relevant in terms of how moral judgments are made: Lau et al. [30] used individualism/collectivism as a concept to compare moral justifications in Canadian and Chinese children aged 7–9 years; Chinese children were more favorable to helping a collective at the expense of harming an individual, whereas Euro-Canadian children showed the reverse pattern.

There are limited studies about moral reasoning in relation to aggression and bullying in South Korea. Studies have examined children or adolescents’ moral reasoning, bullying, or aggressive behavior in relation to their participants’ roles [31], and social-cognitive variables such as empathy, conformity [32], or theory of mind [33]. In Choi and Heo [34]’s study, middle school pupils (13–15 years old) who reported bullying is never acceptable showed higher moral judgment than pupils who indicated that bullying is sometimes acceptable. In particular, half of the pupils reported that social exclusion is acceptable if an excluded child had a fault (e.g., “the child is aggressive, impolite, or a bully”), or is disliked by them. Similarly, Shin and Kim [35] reported that the higher level of aggression was associated with higher levels of moral disengagement in middle school pupils.

Most studies have focused on children or adolescents older than 11 years, and a very limited number of studies exist focused on young children’s moral reasoning about aggressive behavior. Understanding developmental differences in the moral perception of bullying behavior is important for developing effective intervention programs for bullying. Bullying interventions have been heavily focused on the upper grades of elementary school although the age range of elementary school pupils ranges from 6–12 years in South Korea. To date, studies investigating children’s moral reasoning and examining developmental differences are lacking.

### 1.5. Present Study

The great majority of studies on moral reasoning in relation to aggression have been carried out in Western countries. Although these have provided useful information about children’s moral reasoning related to aggressive behavior, there are limitations. Few studies have paid attention to moral reasoning depending on the type of aggression; and of those that have done so, the range of types of aggression examined has been limited. Considering the types of aggression in bullying-like phenomena such as wang-ta in South Korea, it is important to distinguish reasoning about these different types of aggression. This may interact with age and gender. Besides verbal aggression, we distinguished individual and group aggression, as physical group aggression can be one particular feature of wang-ta in the South Korean context. We also included damaging belongings, and separated two kinds of relational aggression (social exclusion; rumor spreading) and two methods of cyber aggression (mobile phone, email).

This study focused on the moral reasoning of children aged 6 and 11 years. The age of 6 years may be seen as a starting point for recognizing bullying-like behaviors in South Korean children [36] while around 11 years is the start of the developmental period during which bullying behavior is most commonly reported and at which negative attitudes towards victims are clearly shown [28,37]. Using cartoon scenarios, we examined four elements of moral reasoning: harmful consequences (would it hurt or not); moral judgment (right or wrong); reason for the judgment (why right/wrong); and causal attribution (why the incident happened). We considered these aspects of moral reasoning in relation to the type of aggression, and age and gender of participants.

Although this research was exploratory given the lack of comparable research in South Korea, we had some expectations. From previous research in Western countries, we anticipated that social exclusion might be seen as less harmful than rumor spreading or physical aggression, but that this might be less applicable in a collectivist country such as South Korea. We also anticipated that older children would judge actions as more wrong, while younger children might take less account of the perceptions and intent of the actors and give more “don’t know” responses for reasons and attributions. Finally, we anticipated that girls would judge actions as more harmful, social exclusion as more wrong, and give more moral domain reasons for judgments than boys.

### 1.6. Aims

The aims of the study were to examine differences in the four aspects of moral reasoning (harmfulness, moral judgment, reason for judgment, causal attribution) by type of aggression, age, and gender, in a South Korean sample.

## 2. Methods

### 2.1. Participants

Elementary school teachers in Incheon, South Korea were contacted by telephone and given a brief explanation of the aims and procedures of the study. Teachers from two elementary schools agreed to participate and the first author visited the schools. Sixty-one children (33 boys, 28 girls) aged 6 years (1^st^ grade) and 96 children (55 boys, 43 girls) aged 11 years (5^th^ grade) participated. The two schools were located close to each other and were matched in the socioeconomic status of their pupil intake (lower–middle class). Thirteen percent (n = 8) of 6 year olds and 57% (n = 55) of 11 year olds had a mobile phone. The data was collected in 2009.

### 2.2. Measures

A cartoon task was used, in which different types of aggressive behavior were depicted [38]. In this study, eight cartoons were used: physical individual (a child hits another child), physical group (several children hit one child), damaging one’s belongings (breaking a ruler), verbal (threats), social exclusion (excluding from play), rumor spreading (telling nasty stories to others), and two types of cyber aggression (sending a nasty text via mobile phone; and sending a nasty message/email via computer).

Each child was interviewed individually in a quiet room in their school. In order to reduce any order-effects of the cartoons shown, the child was first shown the series of all eight figure cartoons before answering any questions. For each cartoon, the behavior displayed was explained by the researcher and the child was asked four questions:

Harmfulness—*Would it hurt the child?* Children were asked to choose on a 4-point scale, 1 (no, it wouldn’t hurt), 2 (not sure), 3 (hurt a bit), 4 (hurt very much).

Moral judgment—*How do you think about this? Do you think this is OK or wrong?* Children were asked to choose on a 4-point scale, 1 (OK), 2 (not sure), 3 (a bit wrong), 4 (very wrong).

Reason for judgment—*Why do you think that?* (open-ended scoring).

Causal attribution—*Why do you think this happened?* (open-ended scoring).

The interviews were conducted individually during school hours and lasted around 15–20 min.

### 2.3. Coding of Open-Ended Questions

Responses for reason for judgment and causal attribution were analyzed using content analysis. Initial categories were developed by the first author from the range of responses obtained, bearing in mind previous categorization schemes by Davidson et al. [7] for reasons for judgment and Smith et al. [9] for causal attributions. These were discussed with two independent researchers. Revised categories were then discussed with the co-authors. If disagreements were found, the categories were further modified and discussed again. This procedure was repeated until the categories were exclusive of each other, and all responses could clearly be assigned to one or more of the categories.

*Reasons for judgment*: Table 1 shows the final 9 categories decided on: “welfare”, “fairness”, “obligation”, “authority and punishment avoidance”, and “prudential reasons” were similar to those used by Davidson et al. [7], and “peer relationships”, “intention”, and “dismiss” were newly established for responses obtained that did not fit into these categories.

*Causal attribution*: Table 2 shows the 4 final categories decided on: for analysis and as some categories had few responses, we coded initial categories of “personal satisfaction”, “characteristic problems”, “practical reasons”, and “disliking victim” into an “aggressor-related” category. Similarly, “provocative victim” and “victim characteristics” were coded into a “victim-related” category. The category “situational factor” was newly established from the responses.

For each open-ended question, responses were coded binomially, with a score of 1 indicating that the category was used and a score of 0 indicating that the category was not used. Most children gave a short response for each open-ended question, that could be assigned to one category. However, if children gave a longer response, that could be assigned to more than one category, then each category was scored. If multiple answers belonged to the same category (e.g., for fun, and venting one’s anger), they were counted as one score (e.g., for aggressor-related; see Table 2).

For each open–ended question, 10 percent of responses were randomly selected and given to the two independent researchers to code, separately. Cohen’s Kappa was calculated and indicated very high reliabilities: *k =* 0.949, *p* < 0.001 for reason for judgment, and *k =* 0.959, *p* < 0.001 for causal attribution.

### 2.4. Statistical Analyses

Scale scores on harmfulness and moral judgment were analyzed by mixed ANOVAs, with type of aggression as a repeated measures factor, and age and gender as independent group factors. Numbers from Huynh–Feldt correction are reported because the sphericity assumption was violated in Mauchly’s test (*p* < 0.05).

For the categorical scores for reason for judgment and causal attribution, Cochran’s Q test was conducted to examine differences in the proportions among all 8 types of aggression, for each category, and then further McNemar tests were conducted for each possible pair by aggression type. Chi-square tests were used to examine age and gender differences.

Due to the large number of comparisons made, we only report results significant at *p* < 0.01 throughout.

### 2.5. Ethical Issues

Verbal consent was obtained from the class teachers and head teachers involved. Generally, parental consent for child participation in research was not required in 2009 in South Korea as long as the children’s teachers agreed. Although negative consequences have been very rare in previous studies [13,28,36,39], the cartoons used in this study might have reminded some participants of their negative experiences related to bullying. If a child did not want to say any more, (s)he was not asked anymore and could stop at any time. Should any child be particularly distressed, the first author had in place an arrangement (agreed with each school) of offering to take them to a teacher or counselor for further support along with the provision of contact details for support networks (e.g., phone number and websites of helplines). However, this did not happen. This study was given approval by the Ethics Committee of the Department of Psychology at Goldsmiths, University of London.

## 3. Results

In general, the children enjoyed the tasks and gave sensible responses. All could answer the questions about harmfulness and moral judgment. For the open-ended questions about reasons and causal attribution, most gave short but sensible replies; usually fewer than one-quarter gave responses which were coded as “don’t know”.

### 3.1. Harmfulness

Table 3 shows mean scores for how harmful pupils judged scenarios to be, by type of aggression, age, and gender. Mainly, children responded that the scenarios hurt very much.

Type of aggression. There was a significant main effect of type of aggression: *F*(6.46, 988.79) = 7.37, *p <* 0.001 with Huynh–Feldt correction (ε = 0.923). Harmfulness scores were highest for physical group aggression and lowest for social exclusion. Pairwise comparisons showed that social exclusion was considered significantly less harmful than physical individual, physical group, and email aggression (*p* < 0.001).

Age. There was a significant main effect of age, *F*(1, 153) = 7.75, *p* < 0.01, partial η^2^ = 0.048. Younger children rated most types of aggression more harmful than older children; this was significant for breaking belongings, *t*(155) = 3.65, *p* < 0.001, and email aggression, *t*(155) = 3.42, *p <* 0.001.

Gender. The main effect of gender was not significant, although girls did give slightly higher scores on all 8 types of aggression. The interaction between gender and type of aggression was not significant.

### 3.2. Moral Judgment

Table 4 shows mean scores for how wrong pupils judged scenarios to be, by type of aggression, age, and gender. Mainly, children responded that the scenarios were very wrong.

Type of aggression. There was a significant main effect for type of aggression, *F*(6.15, 940.64) = 7.93, *p <* 0.001 with Huynh–Feldt correction (ε = 0.88). Wrongness scores were highest for physical group aggression and lowest for social exclusion. Pairwise comparisons showed that physical group aggression was reported as significantly more wrong compared to all other 7 types of aggression (*p <* 0.001).

Age. There was no significant main effect of age.

Gender. There was no significant main effect of gender, although girls did give slightly higher scores on all 8 types of aggression. The interaction between gender and type of aggression was not significant.

### 3.3. Reason for Judgment

The children most commonly used welfare, fairness, and obligation as reasons for their judgment; “don’t know” was also quite commonly scored. The remaining categories (peer relationships, prudential reasons, dismiss, authority, and punishment avoidance and intention) were each mentioned by fewer than 3% of pupils, so they are not included in subsequent analyses. Results for the four categories of welfare, fairness, obligation, and “don’t know”, by type of aggression, age, and gender, are shown in Table 5.

Type of aggression. There were significant differences for welfare, *x*^2^ (7) = 52.86, *p <* 0.001, fairness *x*^2^ (7) = 313.90, *p <* 0.001, obligation, *x*^2^ (7) = 63.57, *p <* 0.001, and “don’t know” *x*^2^ (7) = 46.58, *p <* 0.001. McNemar tests showed that welfare was used significantly more for verbal aggression than physical individual, social exclusion, physical group, and breaking belongings (*p <* 0.01 or *p <* 0.001). It was least commonly mentioned for physical group aggression compared to all other types of aggression except for social exclusion (*p <* 0.01 or *p <* 0.001). Fairness was significantly used more for physical group aggression than social exclusion (*p <* 0.001). Obligation was used significantly more for physical individual aggression, social exclusion, and breaking belongings than for physical group aggression, mobile aggression, and email aggression (all *p <* 0.01). “Don’t know” was used significantly more for email aggression than for physical individual, physical group, verbal, and breaking belongings (all *p <* 0.01).

Age. Older children were more likely than younger children to explain their judgments in terms of the victim’s welfare, or in terms of fairness, whereas younger children were more likely than older children to explain their judgment in terms of obligation. There were significant differences between younger and older children on welfare for verbal (*x*^2^ (1) = 35.36, *p <* 0.001), rumor spreading (*x*^2^ (1) = 10.96, *p <* 0.001), mobile (*x*^2^ (1) = 10.22, *p <* 0.002), and email (*x*^2^(1) = 14.41, *p <* 0.001). Fairness was more likely to be mentioned by older than younger children for physical group aggression (*x*^2^ (1) = 11.40, *p <* 0.001). Obligation was likely to be mentioned by younger than older children for verbal (*x*^2^ (1) = 12.61, *p <* 0.001). Younger children were more likely than older children to have difficulty explaining the reasons for judgment (“don’t know”) for all types, significantly for verbal, rumor spreading, physical group, and email aggression (*p <* 0.01 or *p <* 0.001).

Gender. The main gender difference was in girls using victim’s welfare more in their responses, for all types of aggression except for breaking belongings; this was significant for physical group aggression (*x*^2^ (1) = 11.87, *p <* 0.001), and mobile aggression (*x*^2^(1) = 7.38, *p* = 0.007). Fairness was mentioned at a similar rate between boys and girls except for physical group aggression, which was mentioned significantly more by boys than girls (*x*^2^ (1) = 8.85, *p* = 0.003).

### 3.4. Causal Attribution

Causal attribution for aggressive behaviors was most commonly scored as aggressor-related followed by victim-related, “don’t know”, and situational factor. The percentage of pupils responding in each category, by type of aggression, age, and gender, is shown in Table 6.

Type of aggression. There were significant differences for aggressor-related, *x*^2^ (7) = 77.193, *p* < 0.001; victim-related, *x*^2^ (7) = 63.35, *p* < 0.001; and situational factor, *x*^2^ (7) = 23.73, *p* < 0.01. Aggressor-related was most used for mobile aggression and email aggression, significantly more than for verbal, physical individual, social exclusion, rumor spreading, and physical group aggression (*p* < 0.001 or *p* < 0.01), but breaking belongings showed a significant difference with mobile aggression (*p* < 0.01), not with email aggression.

Aggressor-related was used least in social exclusion than for all other types of aggression; (*p* < 0.001 or *p* < 0.01), but not for physical group aggression. Victim-related was used most for social exclusion and physical group aggression, significantly more than for verbal aggression and breaking belongings (*p* < 0.01 or *p* < 0.001); it was used least for email, mobile, and breaking belongings. Also, physical individual, social exclusion, and physical group aggression were used more than mobile and email aggression (*p* < 0.01 or *p <* 0.001). Rumor spreading was significantly less used than social exclusion and mobile aggression (all *p* < 0.01). Situational factor was mentioned most for social exclusion, significantly more than physical group, mobile, and email aggression (*p* < 0.01 or *p* < 0.001), and more in rumor spreading than email aggression (*p* < 0.01).

“Don’t know” responses, at around one-quarter of total, did not show significant differences by type of aggression.

Age. Aggressor-related was higher in older pupils than younger pupils for all 8 types of aggression, individually significant for rumor spreading (*x*^2^ (1) = 8.54, *p <* 0.01), physical group (*x*^2^ (1) = 8.25, *p <* 0.01), breaking belongings (*x*^2^ (1) = 13.92, *p <* 0.001), mobile (*x*^2^ (1) = 21.71, *p* < 0.001), and email (*x*^2^ (1) = 19.15, *p <* 0.001). Older children also tended to use victim-related attributions more than younger children, except for breaking belongings and mobile aggression, but the age differences were not significant. Differences for situational were inconsistent and not significant. However, younger children made more “don’t know” responses for all 8 types of aggression, and all were significant (*p <* 0.01 or *p <* 0.001).

Gender. No significant gender differences were found.

## 4. Discussion

This study of moral reasoning in 6 and 11 year old South Korean pupils showed that type of aggression is an important factor in the kind of judgments made, reasons given, and causal attribution. Age differences are also apparent; however, gender differences were small. We consider the findings in the light of previous research, and the cultural context in South Korea.

### 4.1. Differences in Moral Reasoning by Type of Aggression

Clear differences in moral reasoning emerged by type of aggression. Physical group aggression was perceived as being the most harmful and the most wrong; social exclusion was perceived as being the least harmful and the least wrong. This is consistent with findings in the US by Goldstein et al. [25] in preschoolers, and Murray-Close et al. [10] (on 4^th^ and 5^th^ graders, so about 9–10 years). Goldstein and Tisak [25] found gossip (rumor spreading) was considered to be more wrong by 12–13 year olds than social exclusion, which is also consistent with the trend of our findings (Table 3), although non-significant.

These findings are noteworthy given the different collectivist context in South Korea compared to the US or other Western countries. Although physical forms of aggression certainly happen, social exclusion is a more salient form of aggression, even captured in the etymology of the term commonly used by pupils, wang-ta [28]. Perhaps because it is relatively common, it is seen as less harmful and wrong (Table 3 and Table 4), is seldom described in terms of the victim’s welfare or rights (fairness) (Table 5), and the victim is relatively often held responsible for what is happening (Table 6). These judgments seem to apply to social exclusion as an indirect form of aggression; actual group physical aggression was described as the most harmful and wrong (Table 3 and Table 4), and as being unfair (Table 5), which may reflect the obviously greater imbalance of power in group physical aggression. However, again the victim is relatively often held responsible for what is happening, at the same level as the aggressor(s) (Table 6) in the aggression-related group of aggressors regardless of the level of harmfulness or wrongness.

Thus, many children in our South Korean sample seem to think of social exclusion as the victim’s responsibility (i.e., victim’s provocation or personal characteristics such as incompetence in play, appearance, or selfishness). Children tend to think that doing wang-ta (social exclusion) is bad but there are some reasons why a child becomes wang-ta [28]. Partly, this is consistent with Park and Killen’s [19] study at 10 and 13 years: children viewed a personality trait as a legitimate basis for peer rejection in terms of making the peer group function well. In their study, findings regarding reasons for peer rejection were rather similar in South Korean and US samples.

For most types of aggression, the aggressor(s) were most frequently held responsible (Table 6); this was particularly the case for cyber aggression, by mobile phones or email. This is consistent with Wright [40] in which aggressor blame was most commonly mentioned by cybervictims as reason for cybervictimization. Sending an aggressive text or email, when not in the same immediate physical and social context as the recipient, would be easiest to attribute to the initiator.

### 4.2. Differences in Moral Reasoning by Age

We did not confirm the expectation that older children would perceive aggression as more wrong than younger children (e.g., [16]); the (non-significant) trend was in the opposite direction (Table 4). A similar trend was seen for scores of how harmful the behavior was, significant for breaking belongings and email aggression which younger children considered as more harmful than did older children. It may be that the younger children were relying more on obligation (in Table 5). Consistent with general theories of moral development [8], older children made more use of the moral domains of reasoning—welfare and fairness—than younger children. Young children tend to respond with stereotypical answers for moral violations such as “it is wrong” [41] rather than explain the reason specifically. It is not clear that this answer comes from either moral or social-conventional reasoning.

This varied by type of aggression; for example, fairness was more common as a reason in older children for physical group aggression in which inequality of power exists. This may be related to the perception of bullying among young children; they were less likely to perceive an imbalance of power in bullying [42].

As expected, younger children made more “don’t know” responses, both for reasons (Table 5) and causal attributions (Table 6) which shows developmental differences to specify their reasoning between young and older children [41].

### 4.3. Differences in Moral Reasoning by Gender

Gender differences were much less prominent than age differences. There were trends for girls to see all types of aggression as more harmful and more wrong (Table 3 and Table 4) but differences were small and did not reach statistical significance. This is consistent with previous studies [15,18]. The reason for gender differences is not clear. From evolutionary psychology, this is sometimes related to the empathic ability of females required for caring for offspring [43].

However, this does not mean that females are morally superior to males. Rather it may reflect different ways of thinking [44] about certain moral violations. The findings of causal attributions of this study also support this. In this study, no gender differences in causal attribution reached significance (Table 6) and two significant differences were found in the reasons given for the judgment (Table 5). In explaining physical group aggression, girls made more use of welfare explanations, whereas boys made more use of fairness explanations. Girls also made more use of welfare explanations for mobile aggression. These gender differences are similar to those reported in some Western studies [21]. This is the same tendency in which females showed consistently higher scores than males on care, fairness, and purity across 67 countries [45].

Gilligan [46] asserts that females’ morality relates to care of others whereas males tend to see morality as justice although there have been controversies about this.

Furthermore, different judgments can be made depending on context or situation of the moral violations [47]. Therefore, the type of aggression is clearly one important factor in examining gender differences in judgments about aggressive behavior. It is necessary to consider different ways of reasoning about aggressive behavior between girls and boys and reflect the difference in developing bullying intervention programs.

## 5. Implications

The results of this study emphasize that examining one or two types of aggression does not adequately show how children think about aggressive behavior. South Korean children clearly distinguished social exclusion and physical group aggression in respect to moral judgment, perception of harmfulness, consideration of the reason for judgment, and causal attribution. Low sensitivity to social exclusion is similar to the attitude towards wang-ta in which the victim is often thought to cause or deserve the victimization. In South Korean schools, moral education has been emphasized, especially since 2011 when several pupils committed suicide after being wang-ta. The South Korean government established a law for character education, requiring all schools and kindergartens to explicitly include elements of moral education (such as consideration, respect, empathy, etc.) in their curriculum [12].

Age-related differences were more prominent than gender differences in moral reasoning. As children grow up, they tend to be more likely to consider moral concepts such as fairness or others’ welfare when judging aggressive behavior. There are certain situations in which those aspects of morality are more frequently considered (e.g., rumor spreading, physical group aggression). Gender differences were small, but girls and boys weighed fairness or other’s welfare rather differently for physical group aggression.

### Table Strengths and Limitations

A strength of this study is that it investigated four elements of moral reasoning; harmfulness, wrongness, justification of judgments, and causal attribution, together. Furthermore, eight types of aggressive behavior were considered. This study provides further understanding related to children’s moral perceptions of aggressive behavior in a collectivist culture. However, some limitations should be noted. Only two schools were sampled, so generalization in this respect is limited. Moral reasoning is only one part of a complex process of moral domain research, which also includes moral emotions, motivation, and identity [2]. In addition, this study did not include the relation between children’s moral reasoning and their real behaviors.

Furthermore, single items were used for each type of aggression in this study. Future research could include more than one example of each type of aggression. In this study, we compared individual and group aggression only for physical aggression, because of the characteristics of wang-ta and previous findings [13,28]. Further research could investigate how differences between individual and group aggression in various formats may lead to a change in moral reasoning.

The causal attribution of social exclusion as often being the victim’s responsibility may reflect the particular nature of wang-ta in South Korea; however, this deserves further investigation in a comparative study with other collectivist and more individualist cultures, to see how specific this is to cultural context.

## Figures and Tables

**Table 1 ijerph-18-02288-t001:** Categories for reason for judgment (Why do you think that?).

Categories	Definition of Categories (with Examples)
1. Welfare	Appeals to victim’s physical, psychological harm, injury, loss, or negative affect (“The child got hurt”)
2. Fairness	Appeals to maintaining a balance of rights between persons (“this is unfair because the child can’t defend himself if several children hit him/her”)
3. Obligation	References to keeping rules or prohibition, or the act is bad itself (“We should not break other’s property” “We should be nice to friends” “Hitting is bad/wrong”)
4. Authority and punishment avoidance	Appeals to the approval of specific authority and punishment of other persons (“My mum said this is bad” “The child would get arrested by policeman”)
5. Peer relationships	Affects peer relationships between children (“Other classmates will get to know about the rumor”)
6. Prudential reasons	Affects personal health, safety, or comfort (“If the child says bad words, he will get used to it” “the child will get a bad habit”)
7. Intention	Intention of act (“It is OK if the child broke the other’s pencil by mistake”).
8. Dismiss	Regard it as not serious or important, or as easy to solve (“It’s OK, you can buy one later”)
9. Don’t know	“I don’t know” or child did not answer

**Table 2 ijerph-18-02288-t002:** Categories for causal attribution (“Why do you think this happened?).

Categories	Definition of Categories (with Examples)
Aggressor related	Personal satisfaction: For fun, boredom, wanting to annoy others
	Characteristic problem: Venting one’s anger, obtaining other’s attention
	Practical reasons: Obtaining resources (“The child is extorting money”, “The child is jealous of the other’s goods”) Disliking victim: Aggressor dislikes a victim or child but did not explain the reason for disliking
Victim related	Provocative victim: Victim provoked first
	Victim characteristics: Caused by victim’s oddness (“The child is ugly and odd”), characteristic problem (“The child is selfish, not kind”), competence (“The child is bad at play”), wang-ta (“The child is wang-ta”)
Situational factor	Attributed to situational factor, rather than to bully or to victim (“There is no room”, “There would be misunderstanding among these children”, “They fought”)
Don’t know	“I don’t know” or child did not answer

**Table 3 ijerph-18-02288-t003:** Means (on 1–4 scale) and standard deviations of harmfulness by type of aggression, age, and gender.

Type of Aggression	Age		Gender		Total
	6 y (n = 61)	11 y (n = 95)	Boy (n = 86)	Girl (n = 71)	N = 156
Verbal	3.66 (0.77)	3.45 (0.84)	3.48 (0.89)	3.59 (0.73)	3.53 (0.82)
Phys. Ind	3.79 (0.49)	3.61 (0.70)	3.63 (0.70)	3.75 (0.53)	3.68 (0.63)
Soc. Excl	3.49 (0.85)	3.35 (0.82)	3.31 (0.90)	3.52 (0.73)	3.41 (0.83)
Rumor	3.52 (0.79)	3.60 (0.69)	3.52 (0.79)	3.63 (0.64)	3.57 (0.73)
Phys. Grp	3.87 (0.39)	3.90 (0.34)	3.87 (0.37)	3.90 (0.35)	3.89 (0.36)
Breaking	3.72 (0.61) **	3.33 (0.71)	3.41 (0.71)	3.58 (0.67)	3.48 (0.70)
Mobile	3.70 (0.72)	3.44 (0.86)	3.49 (0.88)	3.61 (0.73)	3.54 (0.81)
Email	3.85 (0.40) **	3.56 (0.66)	3.63 (0.63)	3.73 (0.53)	3.68 (0.60)
Average	3.71 (0.05)	3.54 (0.04)	3.56 (0.04)	3.68 (0.04)	

** *p* < 0.001; Total: Means and standard deviations of harmfulness for each type of aggression. Average: means and standard deviations of harmfulness averaged across 8 types of aggression. Verbal: Verbal, Phys. Ind: Physical individual, Soc. Excl: Social exclusion, Rumor: Rumor spreading, Phys. Grp: Physical group, Break: Breaking belongings, Mobile: sending a nasty text by mobile, Email: sending a nasty email.

**Table 4 ijerph-18-02288-t004:** Means (on 1–4 scale) and standard deviations for moral judgment (how wrong) by type of aggression, age, and gender.

Type of Aggression	Age		Gender		Total
	6 y (n = 61)	11 y (n = 95)	Boy (n = 86)	Girl (n = 71)	N = 156
Verbal	3.82 (0.50)	3.67 (0.53)	3.69 (0.60)	3.80 (0.40)	3.74 (0.52)
Phys. Ind	3.79 (0.58)	3.70 (0.58)	3.71 (0.63)	3.76 (0.52)_	3.73 (0.52)
Soc. Excl	3.64 (0.68)	3.42 (0.63)	3.44 (0.70)	3.58 (0.60)	3.50 (0.66)
Rumor	3.66 (0.66)	3.79 (0.43)	3.69 (0.62)	3.80 (0.40)	3.74 (0.53)
Phys. Grp	3.90 (0.35)	3.98 (0.14)	3.93 (0.30)	3.97 (0.17)	3.95 (0.25)
Breaking	3.82 (0.43)	3.54 (0.63)	3.59 (0.66)	3.72 (0.45)	3.65 (0.58)
Mobile	3.77 (0.56)	3.66 (0.54)	3.69 (0.58)	3.72 (0.51)	3.70 (0.55)
Email	3.85 (0.40)	3.65 (0.56)	3.70 (0.58)	3.76 (0.43)	3.73 (0.51)
Average	3.78 (0.04)	3.68 (0.03)	3.69 (0.03)	3.78 (0.03)	3.72 (0.28)

Total: Means and standard deviations for each type of aggression. Average: means and standards of wrongness averaged across 8 types of aggression. Verbal: Verbal, Phys. Ind: Physical individual, Soc. Excl: Social exclusion, Rumor: Rumor spreading, Phys. Grp: Physical group, Break: Breaking belongings, Mobile: sending a nasty text by mobile, Email: sending a nasty email.

**Table 5 ijerph-18-02288-t005:** Percentage of children giving four categories for reason for judgment, by type of aggression, age, and gender.

Type of Aggression		Welfare	Fairness	Obligation	Don’t Know
Verbal	Total	38.9	0.6	36.9	17.8
	Age (6/11)	9.8/57.3 **	0/1	54.1/26.0 **	29.7/11.5 *
	Gender (B/G)	34.9/43.7	0/1.4	36.0/38.0	20.9/14.1
Phys. Ind	Total	26.1	1.3	44.6	20.4
	Age (6/11)	16.4/32.3	0/2	54.1/38.5	26.2/16.7
	Gender (B/G)	20.9/32.4	1.2/1	46.5/42.3	24.4/15.5
Soc. Excl	Total	14.0	15.3	41.4	21.0
	Age (6/11)	11.5/15.6	8.2/19.8	49.2/36.5	27.9/16.7
	Gender (B/G)	12.8/15.5	14.0/16.9	45.3/36.6	19.8/22.5
Rumor	Total	29.9	4.5	27.4	29.3
	Age (6/11)	14.8/39.6 *	0/7.3	37.7/20.8	42.6/20.8 *
	Gender (B/G)	26.7/33.8	3.5/5.6	27.9/26.8	31.4/26.8
Phys. Grp	Total	14.6	49.7	21.7	11.5
	Age (6/11)	11.5/16.7	32.8/60.4 *	31.1/15.6	23.0/4.2 *
	Gender (B/G)	5.8/25.4 **	60.5/36.6 **	18.6/25.4	11.6/11.3
Break	Total	19.7	3.2	46.5	15.3
	Age (6/11)	13.1/24.0	0/5.2	50.8/43.8	24.6/9.4 *
	Gender (B/G)	19.8/19.7	4.7/1.4	41.9/52.1	16.3/14.1
Mobile	Total	27.4	3.2	28.0	28.7
	Age (6/11)	13.1/36.5	0/5.2	39.3/20.8	39.3/21.9 *
	Gender (B/G)	18.6/38.0 **	3.5/2.8	30.2/25.4	32.6/23.9
Email	Total	28.7	6.4	20.4	33.1
	Age (6/11)	11.5/39.6 **	0/11.5	29.5/14.6	49.2/22.9
	Gender (B/G)	22.1/36.6	7.0/5.6	25.6/14.1	32.6/33.8
Average	Total Age (6/11) Gender (B/G)	24.9 12.7/32.7 20.2/30.6	10.5 5.1/14.1 11.8/8.9	33.3 43.2/27.1 34.0/32.6	22.1 32.8/15.5 23.7/20.3

** p <* 0.01, ** *p <* 0.001. Average: Means of percentages across 8 types of aggression in each category. Verbal: Verbal, Phys. Ind: Physical individual, Soc. Excl: Social exclusion, Rumor: Rumor spreading, Phys. Grp: Physical group, Break: Breaking belongings, Mobile: sending a nasty text by mobile, Email: sending a nasty email. Total: Proportion of participants who responded in a category.

**Table 6 ijerph-18-02288-t006:** Percentages of children giving each category of causal attribution, by type of aggression, age, and gender.

Type of Aggression		Aggressor Related	Victim Related	Situational	Don’t Know
Verbal	Total	43.3	19.1	7.6	29.9
	Age (6/11)	34.4/49.0	16.4/28.1	1.6/12.5	47.5/18.8 **
	Gender (B/G)	40.7/46.5	22.1/15.5	7.0/8.5	30.2/29.6
Phys. Ind	Total	40.1	26.8	10.2	22.9
	Age (6/11)	29.5/46.9	32.8/33.3	4.9/13.5	36.1/14.6 *
	Gender (B/G)	36.0/45.1	24.4/29.6	15.1/4.2	24.4/21.1
Soc. Excl	Total	25.5	40.1	13.4	21.0
	Age (6/11)	21.3/28.1	31.1/46.9	16.8/13.5	34.1/12.5 **
	Gender (B/G)	19.8/32.4	41.9/38.0	16.3/9.9	22.1/19.7
Rumor	Total	38.9	26.1	10.8	24.2
	Age (6/11)	24.6/47.9 *	27.9/30.2	4.9/3.5	42.6/12.5 **
	Gender (B/G)	33.7/451	26.7/25.4	10.5/11.3	29.1/18.3
Phys. Grp	Total	35.0	35.7	4.5	24.8
	Age (6/11)	21.3/43.8 *	32.8/42.7	4.9/4.2	41.0/14.6 **
	Gender (B/G)	33.7/36.6	40.7/29.6	1.2/8.5	24.4/25.4
Break	Total	46.5	19.7	5.1	28.7
	Age (6/11)	27.9/58.3 **	19.7/18.8	8.2/11.5	42.6/19.8 **
	Gender (B/G)	44.2/49.3	15.1/25.4	7.0/2.8	33.7/45.2
Mobile	Total	60.5	13.4	4.5	21.7
	Age (6/11)	37.7/75.0 **	23.0/15.6	3.3/2.1	37.7/11.5 **
	Gender (B/G)	65.1/54.9	11.6/15.5	2.3/7.0	20.9/22.5
Email	Total	59.2	15.9	2.5	22.3
	Age (6/11)	37.7/72.9 **	18.0/18.8	3.3/3.1	39.3/11.5 **
	Gender (B/G)	64.0/53.5	10.5/22.5	1.2/4.2	24.4/19.7
Average	Total Age (6/11) Gender (B/G)	43.6 29.3/52.7 42.2/45.4	24.6 25.4/24.1 24.1/25.1	7.3 5.1/8.7 7.6/7.0	24.4 40.2/14.5 26.2/22.4

* *p* < 0.01, ** *p* < 0.001 Average: Means of percentages across 8 types of aggression in each category; Verbal: Verbal, Phys. Ind: Physical individual, Soc. Excl: Social exclusion, Rumor: Rumor spreading, Phys. Grp: Physical group, Break: Breaking belongings, Mobile: sending a nasty text by mobile, Email: sending a nasty email.

## Data Availability

Not applicable.

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
