# Peer review of "Moral Reasoning about Aggressive Behavior in Relation to Type of Aggression, Age and Gender in South Korean Pupils"

_ijerph, 2021, doi:10.3390/ijerph18052288_

Round 1

Reviewer 1 Report

Dear authors,

My comments/ suggestions for this paper entitled "Moral reasoning about aggressive behavior in relation to type of aggression, age, and gender in South Korean pupils" are as follows:

  1. The introduction describes the international context scenario; however, there is less information provided on the South Korean context. Even in the 1.4 subheadings "Studies in South Korea," the information provided is limited. Please provide more information about moral reasoning or aggressive behaviors in the South Korean context. 
  2. Was this study based on any health model? 
  3. I understand that data collection was conducted in 2009; however, the ethical consideration has become very strict in recent years. As per the current protocol, students need to provide an assent form or parental consent for children under 18 years. The verbal or written consent taken in this study is not explicit. Please clarify this information in 2.5 Ethical issues.
  4. The discussion seems a little inadequate. Please consider adding more supportive papers and argue your results.
  5.  As per your results, girls gave slightly higher scores on all eight types of aggressions. Although gender differences were less prominent than age differences, please explain why there occurred trends for girls to see all types of aggression more harmful in your discussion section.
  6. Please change your reference style as per the IJERPH instructions.

Reviewer 2 Report

This is a significant contribution to moral reasoning literature.  More nonwestern perspectives are desperately needed.  While complete the tables could be reviewed for minor typos and the significance more clearly represented/identified. Nice work.

Reviewer 3 Report

This study is interesting because it deepens the knowledge about the moral reasoning that children make regarding aggressive behavior.

However, I have noticed some limitations:
- The authors do not specify the type of sampling they used to select the 2 participating schools.
- It is not specified the moment in which the answers were collected (during school hours ...), how long each interview lasted approximately ...
- On the other hand, the authors specify that in 2009 parental authorizations were not requested, does that mean that the data was collected in 2009? If so, I am concerned that the results cannot be extrapolated to the present, since the moral reasoning of children from more than 10 years ago has very likely changed.
- Finally, the citations in the text are not numbered with the numbers that appear in the list of references.

I trust that they can solve my doubts.

Round 2

Reviewer 1 Report

Thank you for revising the paper. I suggest publishing this paper as per journal's protocol.